# Neural network-based method to stratify people at risk for developing diabetic foot: A support system for health professionals

**Ana Cláudia Barbosa Honório Ferreira[1,2], Danton Diego Ferreira[3]\*, Bruno Henrique Groenner Barbosa[3], Uiara Aline de Oliveira[1], Estefânia Aparecida Padua[2], Felipe Oliveira Chiarini[3], Maria Helena Baena de Moraes Lopes[1]**

1 School of Nursing, Universidade Estadual de Campinas, Campinas, São Paulo, Brazil, 2 University Center of Lavras, Unilavras, Lavras, Minas Gerais, Brazil, 3 Automation Department, Universidade Federal de Lavras, Lavras, Minas Gerais, Brazil

\* danton@ufla.br

**Data Availability Statement:** The data file is available from the Mendeley database (https://data.mendeley.com/datasets/hd7wtgs7dz/1). Ferreira,

## Abstract

### Background and objective

Diabetes Mellitus (DM) is a chronic disease with a high worldwide prevalence. Diabetic foot is one of the DM complications and compromises health and quality of life, due to the risk of lower limb amputation. This work aimed to build a risk classification system for the evolution of diabetic foot, using Artificial Neural Networks (ANN).

### Methods

This methodological study used two databases, one for system design (training and validation) containing 250 participants with DM and another for testing, containing 141 participants. Each subject answered a questionnaire with 54 questions about foot care and sociodemographic information. Participants from both databases were classified by specialists as high or low risk for diabetic foot. Supervised ANN (multi-layer Perceptron—MLP) models were exploited and a smartphone app was built. The app returns a personalized report indicating self-care for each user. The System Usability Scale (SUS) was used for the usability evaluation.

### Results

MLP models were built and, based on the principle of parsimony, the simplest model was chosen to be implemented in the application. The model achieved accuracy, sensitivity, specificity, positive predictive value, and negative predictive value of 85%, 76%, 91%, 89%, and 79%, respectively, for the test data. The app presented good usability (93.33 points on a scale from 0 to 100).

### Conclusions

The study showed that the proposed model has satisfactory performance and is simple, considering that it requires only 10 variables. This simplicity facilitates its use by health professionals and patients with diabetes.

Danton; Resende, Igor; Oliveira, Henrique; Ferreira, Ana Claudia; Lopes, Maria Helena; Anjos, Andre (2022), "Data for: Competitive Neural Layer-based Method to Identify People with High Risk for Diabetic Foot", Mendeley Data, V1, doi: 10.17632/hd7wtgs7dz.1.

**Funding:** The authors received no specific funding for this work.

**Competing interests:** The authors have declared that no competing interests exist.

# 1. Introduction

Diabetes Mellitus (DM) and its complications are becoming one of the biggest causes of morbidity and mortality in the world. In 2017, 4 million people died from diabetes and its complications. In 2019 there were 463 million people, between 20 and 79 years of age, diagnosed with DM worldwide, with a projected increase of 51% by 2045 [1, 2].

Neuropathy is a complication often found in people with diabetes, which compromises the functioning of nerves throughout the body and alters sensory, motor, and autonomic functions. The presence of neuropathy makes the person even more prone to the development of diabetic foot, especially due to the loss of sensation in the feet [3, 4]. Diabetic foot is a complication that affects up to 25% of people with diabetes throughout their lives and is a serious health problem among chronic patients, possibly leading to lower limb amputations. It represents a relevant socioeconomic impact due to the loss of ability to work, affects social life and the quality of life, with high morbidity, disability, and mortality, and can be classified as one of the most devastating complications of diabetes [1, 4–6].

The International Diabetes Federation argues that a greater focus on the prevention of foot ulcers of people with diabetes is needed, rather than focusing on treating them, as there is enormous difficulty in treating and recuperating the limb after the injury has already been established [1]. For the prevention of diabetic foot, it is usually necessary to screen people with diabetes, through clinical examination, which generates a risk score for the patient to develop the diabetic foot [7, 8]. Screening requires time, knowledge, training, and skill from the professional (nurse or doctor) for an effective result. Furthermore, there are a lot of services that need to be performed daily by the health unit professionals [9], which makes it difficult to examine all people with diabetes [10]. To prioritize the clinical examination of those at most risk for developing this complication, alternative methods are needed to quickly identify, people more susceptible to develop diabetic foot.

Artificial intelligence (AI) has been widely used in the health area to assist health professionals in decision making with a focus on screening, classification, and diagnostics, among other possibilities. The work reported in [11] developed a computer vision approach for the classification of skin cancer using hybrid texture features. They achieved an overall accuracy of 97.13% using a multi-layer perceptron (MLP). An MLP is an ANN capable of handling both linearly separable and non-linearly separable data. It belongs to a class of neural networks known as feed-forward neural networks, which connect neurons in one layer to the next layer in a forward manner. The interconnected neurons process data through three or more layers. The basic structure of an MLP consists of an input layer, one or more hidden layers and an output layer. Due to their simplicity, MLPs usually require short training times to learn the representations in data and to produce an output. Alternatively, deep neural networks (DNNs), which are artificial neural networks with deep layers, have also been employed for developing health decision making systems. Due to the automatic feature extraction ability, deep learning methods have been successfully applied in different areas, especially in the field of medical imaging [12]. To investigate spine pathologies, the authors of the work reported in [13] proposed deep learning primitives and stacked Sparse autoencoder-based patch classification modeling for Vertebrae segmentation from Computed Tomography (CT) images. A deep learning approach was also developed in [14] for automatic CT vertebra segmentation, which significantly reduced computational cost and achieved excellent accuracy in vertebra segmentation. The Convolutional Neural Network, which is a type of DNN, was employed for Liver Segmentation in Medical Diagnosis [15]. Liver segmentation in CT scan images is a significant step toward the development of a quantitative biomarker for computer-aided diagnosis [16]. In the work reported in [17], the Convolutional Neural Network (CNN) and Variational

Autoencoder deep learning models were used for feature extraction to build a binary and multiclass classification system for supporting COVID-19 triage based on computed tomography data. Despite their ability to deal with complex problems, DNNs usually require longer training periods of time. Additionally, they rely on powerful computers with specialized processing units such as Tensor Processing Units (TPU) and Neural Processing Units (NPU).

Regarding diabetic foot, some AI-based methods have been developed to identify people at risk for diabetic foot [18–22] while others focus on changes already present in feet of people with DM [23–28]. In the research presented in [19] the authors used a questionnaire and investigated the presence of risk factors for diabetic foot. Data analysis was performed using Self-Organizing Maps (SOM), which led to the identification of two different groups of participants. A high-risk group was identified based on having the highest number of risk factors present among the participants. In the research reported in [20], the authors used the same database introduced in [19]; however, they employed a competitive neural layer (CNL) with only two neurons for the separation of groups. The CNL approach outperformed the SOM approach, with this finding being confirmed by specialists in diabetic foot and statistical tests.

All the aforementioned AI-based works used unsupervised algorithms. Unlike these, the present study aimed to build a supervised ANN for the identification of people at risk for diabetic foot. For the construction of this new model, a database with the information of participants with diabetes was labeled according to the risk for diabetic foot (high or low risk) by diabetic foot specialists, which guided the supervised learning of the ANN. In addition, a further study was carried out on the available variables to reduce the high risk for diabetic foot contributory factors of high risk for diabetic foot and make the data collection more efficient for the end-user.

The primary goal of the proposed risk classifier is to identify people with diabetes who require immediate intervention. By doing so, health professionals can anticipate and prioritize their care. This early evaluation followed by a clinical examination of the feet may direct personalized actions that could help prevent diabetic foot.

## 2. Database

To design (training and validation) the proposed method, a database consisting of 54 variables identified in 250 people with diabetes was used. This database is presented in [20]. These data were collected via questionnaire in a health institution of a city in the state of Minas Gerais, Brazil, from people aged 18 years or over, who agreed to participate in the research study and signed the consent form. The anonymity of the participants' data was ensured, and the data collection started only after authorization from the institution (CAAE 66815617.3.00005404).

The participants were chosen by convenience sampling. The instrument (questionnaire) used for data collection covered the risk factors for the development of diabetic foot proposed by the International Consensus on Diabetic Foot [29], and by the Primary Care Booklet to Assist People with DM from the Brazilian Ministry of Health [30], which resulted in a total of 54 variables. Variables related to people's self-care habits concerning their health, foot care, and perceived changes in the feet, as well as sociodemographic and socioeconomic variables were included. As detailed in [20], each entry in the dataset contained the subject's income, profession, gender, age, years of education, marital status, body mass index (BMI), smoking habit, alcohol use, type of diabetes, presence of hypertension, circulatory problems, burning sensation in the feet, presence of bunions, examination of the feet, wrinkles between the toes, habit of washing the feet, calluses on the feet, type of footwear, time between buying shoes, walking habits, blood glucose value, whether blood glucose is monitored, food control, current or previous ringworm in the feet, pain in the legs and feet, current or previous cracks in the

feet, current or previous blister on the feet, shock in feet, warm and reddish feet, removal of cuticles from toes, type of toenail cutting, sock type, action taken when diabetic foot occurred, checking inside the shoe, position when watching television/reading, length of time standing or walking, type of treatment, length of time with diabetes, practice of physical activity, tingling in the feet, loss of foot sensitivity, numbness in the feet, sitting on the feet, edema in the feet, change in foot structure, moisturizing of the feet, vision problems, current or previous ingrown toenails, use of warm bag on feet, current or previous foot injuries, and the presence of leg/foot amputation. In total, each sample (patient input) in the database contained 54 variables. The data were encoded in which the categorical variables were transformed into floating-point numbers -1.0 (meaning that the value presents lower risk for the diabetic foot condition) or +1.0 (meaning that the value presents higher risk for the diabetic foot condition) [20].

The data of each participant was analyzed, independently, by two diabetic foot experts that classified each person as low or high risk for diabetic foot. A third expert analyzed the discordant cases to obtain a final consensus. Based on their professional experiences and the literature, the experts indicated the risk (high risk or low risk) that each of the participants had for the development of diabetic foot. In total, 169 people were identified by the experts as being at high risk for diabetic foot development and 81 at low risk.

To test the designed method, a new database was constructed. This test database was composed of 141 participants, according to the sample calculation for finite populations [31]. This sample included people, 18 years of age or more, with diabetes registered in a Primary Health Unit (PHU) with the Family Health Strategy Team (Brazilian mode of health care) of a city in Minas Gerais, Brazil, who agreed to participate in the research by signing the consent form. Data collection started after authorization from the institution (CAAE 66815617.3.00005404).

For data collection, a computer program was built with all the questions in the questionnaire. The literal information was automatically coded by the program. After data collection, the information of each participant was analyzed individually by the three experts in diabetic foot. The experts indicated the risk that the 141 participants had for the development of diabetic foot, classifying them as high or low risk. The data were presented to the supervised ANN for its final performance evaluation.

## 3. Data normalization

Data normalization is important to avoid polarization in the training of the neural network [32]. The non-categorical variables of both databases were standardized within the range [−1 1] according to:

$$x_n = \frac{2(x - x_{min})}{(x_{max} - x_{min})} - 1 \tag{1}$$

where **x** is the database variable to be normalized and $x_{min}$ and $x_{max}$ are the maximum and minimum values of the corresponding variable.

## 4. Method design

The design steps of the proposed method are presented by the block diagram shown in Fig 1.

The data were randomly divided into 10 folds of 25 patients each (totaling 250), following the Cross-validation methodology [33].

The use of a multilayer perceptron (MLP) neural network was proposed, in which three approaches were developed (I, II, and III):

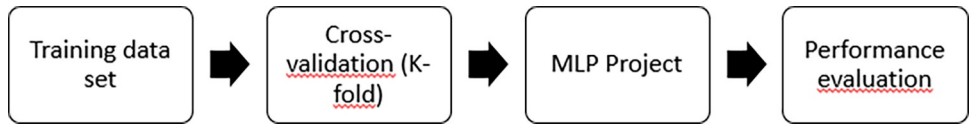

**Fig 1. Proposed method design.**

I—Approach that does not require a variable selection step. In this case, all 54 variables are used by the MLP model;

II—Approach that uses as inputs only the variables identified as most relevant through multivariate analysis, as presented in [20];

III—Approach that uses as inputs only variables selected by the Fisher's Discrimination Ratio (FDR) [34];

In summary, the input of the MLP for approach I is a vector with 54 values corresponding to the 54 variables described in Section 2, after data codification and normalization. Approaches II and III used fewer variables (only the selected ones) than approach I as input for the MLP, due to the specific feature selection step employed by each approach.

The MLP models were configured with an intermediate layer, in which the number of neurons was chosen experimentally, and the logarithmic sigmoid activation function was used. The output layer was composed of two neurons (each one representing a class: high risk or low risk), the target was defined as [1 0] for high risk and [0 1] for low risk. The softmax activation function was used in the output layer since it normalizes the neurons' output to a probability distribution.

The training algorithm was the Bayesian regularization backpropagation [35]. This minimizes the linear combination of the square of errors and weights, determining the best combination that produces a network that generalizes well. This algorithm updates the weights according to the Levenberg-Marquardt optimization algorithm.

As performance measures of the models, accuracy, sensitivity, specificity, positive predictive value (PPV), and negative predictive value (NPV) were calculated. To compare the generated models, the Friedman tests and the Dunn-Bonferroni post-test [36] were applied. The analysis was performed using the statistical software SAS version 9.4 and SPSS version 23, considering a significance level of 5%.

Finally, an app, called CARPeDia, implementing the best MLP model achieved, was built, providing a simple and easy-to-use interface. The app was designed by using the JavaScript, Hypertext Markup Language (HTML) and Cascading Style Sheets (CSS) languages. The System Usability Scale (SUS) [37] was employed for the usability evaluation of the developed app.

## 5. Results

This section presents the results for both the validation and testing datasets.

### 5.1 Results for validation data

Mean accuracy values over the validation data of the 10 folds used during the models' training are presented.

For approach I, the number of neurons in the intermediate layer was varied from 5 to 30 and the best results were obtained with 10 neurons. Therefore, the final model for this approach (MI) had 54 entries, referring to all 54 variables in the database, 10 neurons in the middle layer, and 2 output neurons, that is, 54x10x2.

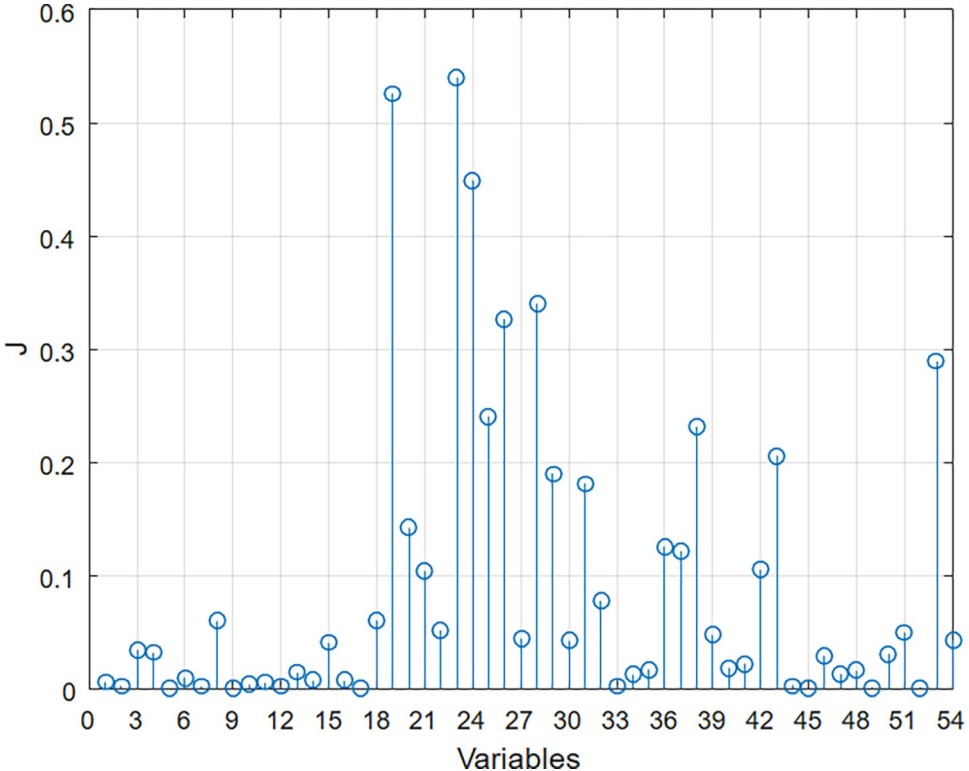

**Fig 2. Result of FDR (Fisher's Discrimination Ratio) for the 54 variables in the database.**

For approach II, a model based on 29 database variables (MII) was proposed, these being the variables indicated as the most relevant in the study reported in [20]. The final configuration of the best model architecture was 29x10x2.

For approach III (MIII), the FDR was applied to select the most relevant variables, that is, those that most contribute to the classifier's performance. The FDR result is shown in Fig 2, where variables with the highest **J** value are considered the most relevant.

From this result, three models were developed in approach III, namely:

- MIII1: considers the 10 most relevant variables, being represented by the following questions (in order of relevance): 23-Do you feel tingling in your legs and feet?; 19- Have you ever felt a loss of sensation in your feet?; 24-Do you feel numbness in your legs and feet?; 28-Do you feel stinging in your legs and feet?; 26-Do you feel shocks in your legs and feet?; 53-Do you have/have you had any sores and/ or bruises on your feet?; 25-Do you feel a burning sensation in your legs and feet?; 38- Do you have/have you had ingrown toenails?; 43- Do you use or have you used a hot water bottle on your feet, and/or soaked your feet in hot water?; 29-Have you noticed any difference in your foot bones? Fig 3 demonstrates the distributions of the answers to these questions over the groups.

- MIII2: considers the 20 most relevant variables—including the previous ten, and

- MIII3: considers the 30 most relevant variables—including the previous twenty.

The number of neurons in the middle layer was varied from 5 to 30 and the final configurations for each model were 10x10x2 (MIII1), 20x20x2 (MIII2), and 30x20x2 (MIII3).

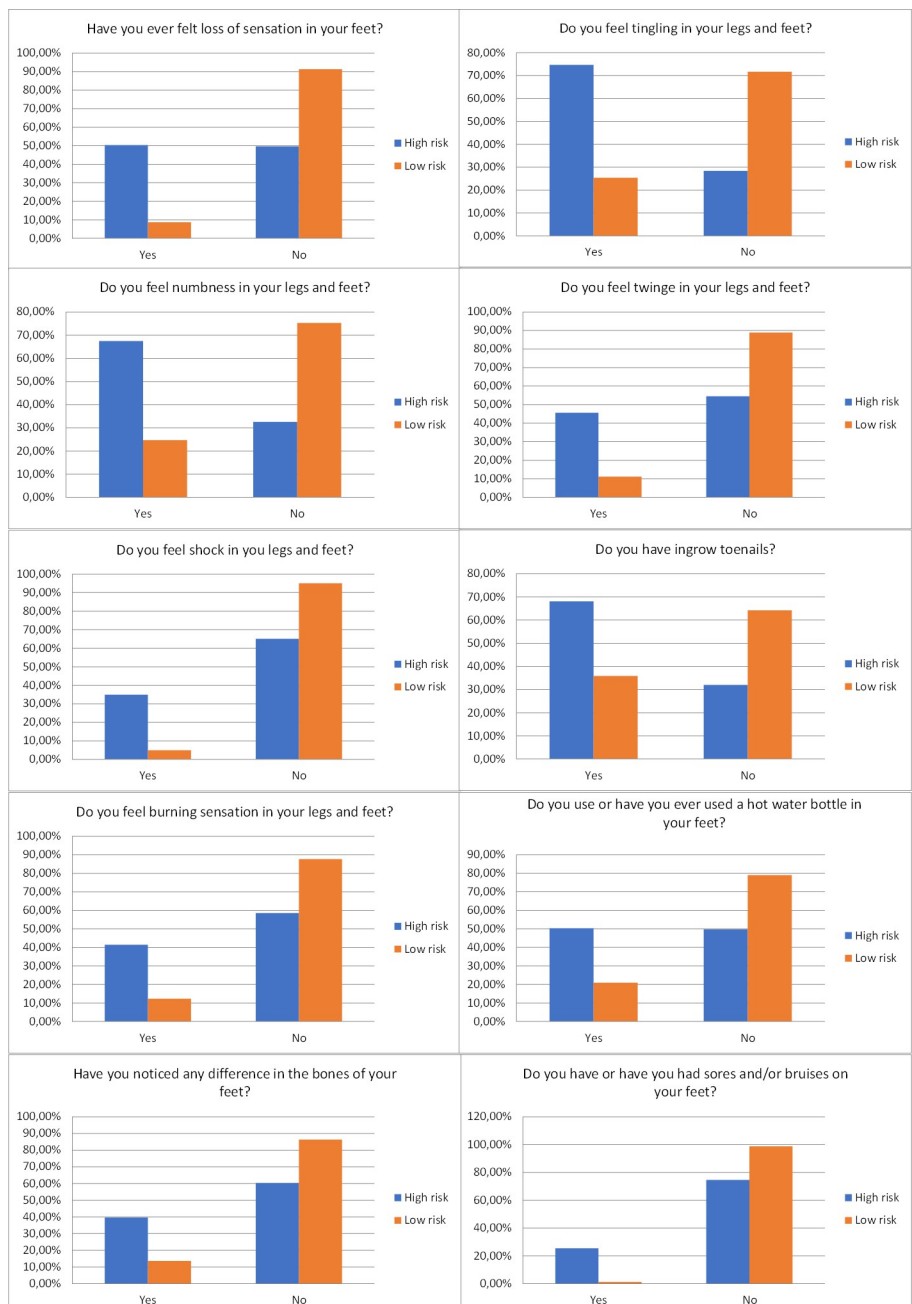

**Fig 3. Frequency of the 10 most discriminating variables, according to the risk for diabetic foot (high or low risk).**

The mean values of the 10 folds of these models for sensitivity, specificity, accuracy, PPV, and NPV were compared and are shown in Table 1. Note that there were no statistically significant differences between the models. Considering the principle of parsimony [38], it was conclude that for approach III, the MIII1 model (10x10x2—with ten variables at the entrance) was the best model to be used.

Models MI, MII and MIII1 were compared with the results shown in Table 2. A significant difference in relation to the PPV in the Friedman test was found, however, the Dunn-Bonferroni post-test showed that there was no significant difference between the models. Therefore,

**Table 1. Comparison among approach III models.**

| Index | Model | Mean ± Standard deviation | Minimum | Q1 | Median | Q3 | Maximum | *p*-value* |
|---|---|---|---|---|---|---|---|---|
| **Sensitivity** | MIII1 | 0,88 ±0,08 | 0,76 | 0,82 | 0,88 | 0,94 | 1,00 | |
| | MIII2 | 0,88 ±0,11 | 0,76 | 0,76 | 0,88 | 1,00 | 1,00 | 0,6065 |
| | MIII3 | 0,86 ±0,12 | 0,71 | 0,71 | 0,85 | 1,00 | 1,00 | |
| **Specificity** | MIII1 | 0,78 ±0,14 | 0,63 | 0,67 | 0,75 | 0,88 | 1,00 | |
| | MIII2 | 0,74 ±0,19 | 0,38 | 0,67 | 0,75 | 0,88 | 1,00 | 0,4493 |
| | MIII3 | 0,76 ±0,16 | 0,50 | 0,63 | 0,75 | 0,88 | 1,00 | |
| **Accuracy** | MIII1 | 0,83 ±0,08 | 0,72 | 0,77 | 0,81 | 0,88 | 0,97 | |
| | MIII2 | 0,81 ±0,13 | 0,60 | 0,76 | 0,81 | 0,94 | 0,97 | 0,8089 |
| | MIII3 | 0,81 ±0,13 | 0,66 | 0,67 | 0,81 | 0,88 | 1,00 | |
| **PPV** | MIII1 | 0,81 ±0,11 | 0,69 | 0,72 | 0,79 | 0,86 | 1,00 | |
| | MIII2 | 0,79 ±0,13 | 0,57 | 0,74 | 0,79 | 0,89 | 1,00 | 0,5968 |
| | MIII3 | 0,79 ±0,14 | 0,62 | 0,65 | 0,76 | 0,87 | 1,00 | |
| **NPV** | MIII1 | 0,87 ±0,08 | 0,78 | 0,81 | 0,84 | 0,94 | 1,00 | |
| | MIII2 | 0,86 ±0,13 | 0,68 | 0,76 | 0,86 | 1,00 | 1,00 | 0,8669 |
| | MIII3 | 0,84 ±0,13 | 0,68 | 0,72 | 0,85 | 1,00 | 1,00 | |

*\**p*-value obtained through the Friedman test.*

as the model MIII1 was the simplest architecture, 10x10x2, and following the principle of parsimony, this was the most suitable.

## 5.2 Results for testing data

This section presents the results of the best model (approach III—MIII1) for the testing database (141 patients). The results are shown in Table 3. The result of the competitive network, developed in [20] was also presented, for the same testing database, for comparison purposes. It was observed that the MIII1 stood out for obtaining the best accuracy, best sensitivity, and

**Table 2. Comparison between models MI, MII and MIII1.**

| Variable | Model | Mean ± Standard deviation | Minimum | Q1 | Median | Q3 | Maximum | *p*-value* |
|---|---|---|---|---|---|---|---|---|
| **Sensitivity** | MI | 0,85 ±0,13 | 0,71 | 0,71 | 0,85 | 1,00 | 1,00 | |
| | MII | 0,89 ±0,07 | 0,82 | 0,82 | 0,88 | 0,94 | 1,00 | 0,5647 |
| | MIII1 | 0,88 ±0,08 | 0,76 | 0,82 | 0,88 | 0,94 | 1,00 | |
| **Specificity** | MI | 0,73 ±0,18 | 0,50 | 0,56 | 0,75 | 0,88 | 1,00 | |
| | MII | 0,77 ±0,19 | 0,50 | 0,63 | 0,71 | 1,00 | 1,00 | 0,2122 |
| | MIII1 | 0,78 ±0,14 | 0,63 | 0,67 | 0,75 | 0,88 | 1,00 | |
| **Accuracy** | MI | 0,79 ±0,14 | 0,60 | 0,67 | 0,79 | 0,94 | 0,97 | |
| | MII | 0,83 ±0,12 | 0,66 | 0,72 | 0,81 | 0,94 | 1,00 | 0,1046 |
| | MIII1 | 0,83 ±0,08 | 0,72 | 0,77 | 0,81 | 0,88 | 0,97 | |
| **PPV** | MI | 0,77 ±0,14 | 0,59 | 0,65 | 0,77 | 0,89 | 1,00 | |
| | MII | 0,81 ±0,15 | 0,62 | 0,69 | 0,76 | 1,00 | 1,00 | **0,0297** |
| | MIII1 | 0,81 ±0,11 | 0,69 | 0,72 | 0,79 | 0,86 | 1,00 | |
| **NPV** | MI | 0,83 ±0,15 | 0,63 | 0,68 | 0,85 | 1,00 | 1,00 | |
| | MII | 0,87 ±0,09 | 0,74 | 0,78 | 0,89 | 0,94 | 1,00 | 0,4966 |
| | MIII1 | 0,87 ±0,08 | 0,78 | 0,81 | 0,84 | 0,94 | 1,00 | |

*\* *p*-value obtained through the Friedman test.*

**Table 3. Testing results (Performance evaluation) of the proposed MIII1 model and the CNL model developed in [20].**

| Model | Accuracy | Sensitivity | Specificity | PPV | NPV |
|---|---|---|---|---|---|
| Competitive network (CNL) | 0,83 | 0,76 | 0,91 | 0,89 | 0,79 |
| MIII1 | 0,85 | 0,84 | 0,89 | 0,86 | 0,85 |

best NPV. Also, using a questionnaire with only 10 questions instead of 54 is preferable for a risk stratification system since it takes less time to apply.

## 6. Discussion

The high sensitivity of the system is important to identify those patients who are at high risk for diabetic foot, helping healthcare professionals to identify those who need immediate care.

The high specificity of the system also demonstrates that people with DM, when using the system and finding a result that confirms that they are not at high risk for diabetic foot, will be able to continue to monitor the health of their feet at home. They can do this without seeking healthcare service, either by using the system on their own cell phone or through a health agent who can apply the system to the patient during home visits. The high PPV and NPV values found confirm the strong capacity of the proposed system.

When analyzing the variables that were most discriminating for the classification of participants according to approach III, it was observed that they are constantly presented in the literature as risk variables for diabetic foot [1, 4, 30]. There is a strong relationship between the variables and the triad of risk factors for ulceration: neuropathy, deformity, and trauma. All ten variables indicated by the approach III model are related to the triad, which reinforces the system's ability to search for the characteristics of greatest risk for ulceration [4, 39].

The complication is characterized as a neuropathic foot when the presence of changes in the sensitivity of the feet is perceived, which often comes with sensations such as tingling, claudication, numbness, burning, and shocks [4]. Once present, the loss of protective sensitivity is characterized as a strong point for the appearance of a foot injury, since, when injured, the person will not notice the pain, and will not treat the injury. Fig 3 shows that these sensations are less perceived by the participants who were classified as low risk for diabetic foot. Conversely, the presence of these sensations was frequently mentioned by the participants in the group classified as high risk.

With the loss of sensation, factors such as the presence of ingrown toenails, and the use of hot water on the feet can be considered precursor factors for trauma, which can cause great tissue damage in the feet [2, 13].

**Table 4. Machine learning-based methods used for diabetic foot risk classification.**

| Reference | Feature Selection | Classifier | Sample Size | Exams | Number of Variables | Number of features selected | Performance |
|---|---|---|---|---|---|---|---|
| [20] | N/A | LR and Random Forest (RF) | 246,705 | Yes | 18 | N/A | 95.00% (Accuracy) |
| [42] | Not informed | LR | 80 | Yes | Not informed | 4 | 83.00% (Accuracy) |
| [43] | Univariate logistic regression | Multivariate LR | 853 | Yes | 40 | 12 | 0.84 (concordance index) |
| [44] | Cox proportional hazard multiple regression analysis | Cox proportional hazard multiple regression analysis | 1810 | Yes | 28 | 6 | 0.62 (AUC) |
| Proposed Method | Multivariation analysis Fisher's discriminant ratio | MLP | 391 | No | 54 | 10 | 85.00% (Accuracy) |

In addition to these findings, it is also possible to perceive the atrophy of the interosseous musculature, enlargement of the plantar arch and "claw" toes, which can characterize the observation of the participants regarding the "difference in the bones of the feet", that is, the deformity. Areas of the foot that experience high pressure are more prone to the appearance of injuries, and tissue damage. Therefore, it is important to identify these areas early in order to provide adequate protection, with adequate shoes and equipment that provide comfort and support for the feet [4, 39].

The presence of foot injury has been considered by the literature as a risk factor for diabetic foot, because once the patient had an injury, they are vulnerable to new occurrences. The patient should investigate the causes of the current and/or previous injury and take step to prevent its recurrence [13, 39].

The presence of characteristics considered to be risk factors for diabetic foot was lower in the group classified as low risk, mainly when the ten most relevant variables (Fig 3) were considered. This corroborates the results achieved by approach III (MIII1), which presented high specificity. Once all these risk characteristics, or some of them, are present in the individual and are identified by the system, a risk score for diabetic foot may be created [4, 30].

Regarding the use of machine learning techniques to assess the risk of developing diabetic foot, recent research has focused on capturing the images of the feet for processing [40, 41] and/or clinical exams [20, 41–44]. The work reported in [40] compared traditional classifiers, such as MLP and Rando Forest using different feature selection and optimization techniques with several state-of-the-art Convolutional Neural Networks (CNNs) on foot thermogram images to identify diabetic foot. They found that the AdaBoost Classifier achieved an F1 score of 97% by processing ten features extracted from foot thermogram images achieving better results than the CNN-based classifiers. The authors of the work reported in [41] used a real-time Single Snapshot Multiple-frequency Demodulation—Spatial Frequency Domain Imaging (SSMD-SFDI) system to map the optical properties of the forefoot skin. They used Logistic Regression and Support Vector Machines (SVM) to process the 2D maps of optical and physiological parameters, including cutaneous hemoglobin concentration, oxygen saturation, scattering properties, melanin content, and epidermal thickness, from every single snapshot. An area under the ROC curve (AUC) of 97.2% was achieved (distinguishing the diabetic foot patients from all other subjects). Despite the high performance achieved by the image-based classifiers, image processing by Deep Neural Networks usually requires lengthy periods of time for training. Additionally, it requires powerful computers with specialized processing units such as Tensor Processing Units (TPU) and Neural Processing Units (NPU). In some cases, a strict experimental setup with specific illumination to capture the images of the feet, which may decrease the usability of the system, must be used as in the SSMD-SFDI imaging system reported in [41].

In relation to machine learning-based systems that avoid the use of images for diabetic foot development risk classification, Table 4 summarizes the main features of four existing methods and the proposed one. Most papers exploited feature selection methods to reduce the number of variables. Logistic Regression (LR) was the classifier most used. The size of the sample studied varied from one work to another. It should be highlighted that the proposed method is the only one that does not require clinical exams to perform the risk classification, which makes the method easier and cheaper for the user. The number of variables varied from 18 to 54 and the final number of variables (after feature selection) varied from 4 to 12. The studies normally used accuracy as the performance metric. It is emphasized that the proposed method achieved a competitive accuracy (85.00%) in comparison to the other studies shown in Table 4, mainly considering that it does not require any further clinical exams to perform the final risk classification.

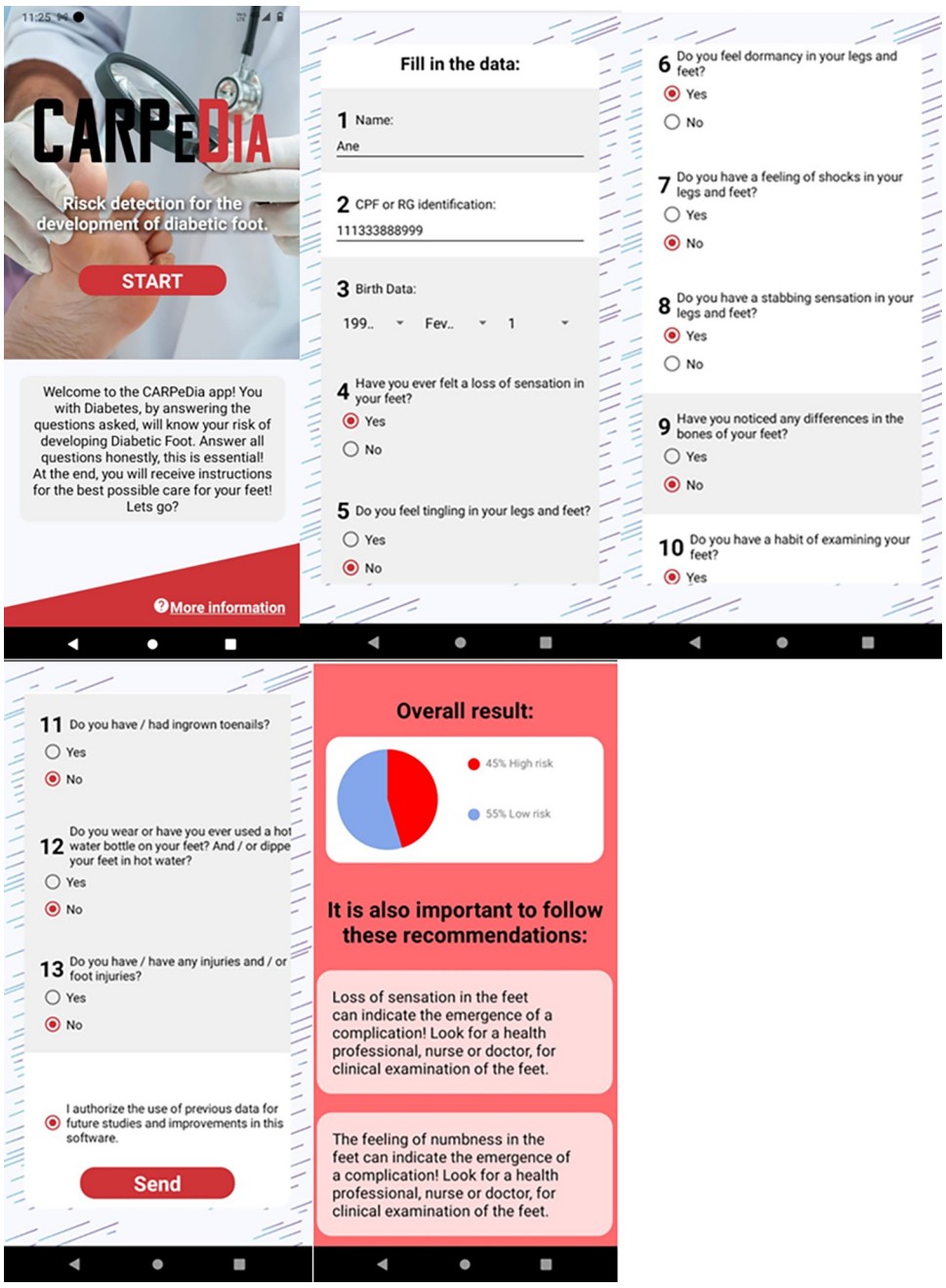

**Fig 4. Example of the CARPeDia app interfaces considering a fictitious patient.**

## 6.1 The CARPeDia app

The use of apps for smartphones has been widely experienced in the health field due to the following reasons: (i) they enable the dissemination and updating of knowledge; (ii) they support the clinical decision-making of professionals in the area, contributing to the elaboration of reliable diagnoses and qualified therapeutic guidelines/conducts for patients/users; and (iii) they can optimize the work of health professionals, particularly in situations with a high number of patients to be treated.

In addition to appealing to those who are able to access it, seeking to monitor and verify the risk for diabetic foot, the CARPeDia also generates a personalized report. This report highlights any incorrect self-care actions performed by the user, and provides guidelines for improving their health and foot care.

The app developed implements the proposed approach III (MIII1), which uses only 10 input variables to stratify the risk for diabetic foot, making the app simpler to be used. The result obtained after the usability evaluation is an index of user satisfaction (ranging from 0 to 100). Following the calculations proposed in [37], a score of 93.33 points was achieved, which demonstrates that the app fulfils the usability requirements.

Fig 4 shows the CARPeDia app interfaces. After completing the information requested in the app, a personalized report is generated, which provides the user guidelines and suggests behavior/attitudes changes. It also guides the changes already perceived by the patient, and addresses aspects that could increase the risk for diabetic foot. Furthermore, the collected information is stored for future analysis and improvements in the method.

## 7. Conclusion

This work presents a new supervised neural network model for diabetic foot risk assessment. The results showed that with only 10 variables out of 54 in the database, the proposed supervised model achieved an accuracy of 85% and a sensitivity of 84%, outperforming the unsupervised model presented in [20]. The dimension reduction from 54 to 10 implies lower computational cost, which is desirable for implementation considering hardware requirements, and also requires less time for patient information collection.

Given the high number of people with DM and the high demand for health services, the system helps by stratifying cases, so that health professionals can organize the flow of care according to priorities, immediately attending to those in need (greater risk).

For future works, the authors intend to exploit Deep Neural Network models [12, 45] to segment and process images of the feet for patients classified as high risk for diabetic foot.

## Acknowledgments

The authors thank John Brooke, the author of the SUS scale, which has been made freely available for use in usability assessments. We also acknowledge the National Council for Scientific and Technological Development (CNPq/Brazil, Proc 306262/2017-7) for supporting this work.

## Ethics statement

All participants gave their informed consent prior to their inclusion in the study. Patient anonymity was preserved. Data collection started after authorization from the Research Ethics Committee of the institution (CAAE 66815617.3.00005404 and CAAE 66815617.3.00005404).

## Author Contributions

**Conceptualization:** Danton Diego Ferreira, Maria Helena Baena de Moraes Lopes.

**Data curation:** Ana Cláudia Barbosa Honório Ferreira, Uiara Aline de Oliveira, Estefânia Aparecida Padua.

**Formal analysis:** Danton Diego Ferreira, Bruno Henrique Groenner Barbosa, Maria Helena Baena de Moraes Lopes.

**Investigation:** Ana Cláudia Barbosa Honório Ferreira, Danton Diego Ferreira, Uiara Aline de Oliveira, Maria Helena Baena de Moraes Lopes.

**Methodology:** Ana Cláudia Barbosa Honório Ferreira, Danton Diego Ferreira, Bruno Henrique Groenner Barbosa, Uiara Aline de Oliveira, Estefânia Aparecida Padua, Felipe Oliveira Chiarini, Maria Helena Baena de Moraes Lopes.

**Software:** Danton Diego Ferreira, Bruno Henrique Groenner Barbosa, Felipe Oliveira Chiarini.

**Supervision:** Maria Helena Baena de Moraes Lopes.

**Validation:** Ana Cláudia Barbosa Honório Ferreira.

**Writing – original draft:** Ana Cláudia Barbosa Honório Ferreira.

**Writing – review & editing:** Danton Diego Ferreira, Bruno Henrique Groenner Barbosa, Maria Helena Baena de Moraes Lopes.

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
