## [Decision Letter · Decision Letter 0]

13 Dec 2022

PONE-D-22-26803Neural network-based method to stratify people at risk for developing diabetic foot: a support system for health professionalsPLOS ONE

Dear Dr. Ferreira,

Thank you for submitting your manuscript to PLOS ONE. After careful consideration, we feel that it has merit but does not fully meet PLOS ONE’s publication criteria as it currently stands. Therefore, we invite you to submit a revised version of the manuscript that addresses the points raised during the review process.

We look forward to receiving your revised manuscript.

Kind regards,

Imran Ashraf, Ph.D.

Academic Editor

PLOS ONE

Journal Requirements:

5. Please upload a new copy of Figure xxxx as the detail is not clear. Please follow the link for more information: " ext-link-type="uri" xlink:type="simple">https://blogs.plos.org/plos/2019/06/looking-good-tips-for-creating-your-plos-figures-graphics/"
https://blogs.plos.org/plos/2019/06/looking-good-tips-for-creating-your-plos-figures-graphics

Reviewers' comments:

Reviewer's Responses to Questions

**Comments to the Author**

1. Is the manuscript technically sound, and do the data support the conclusions?

Reviewer #1: Yes

Reviewer #2: Partly

2. Has the statistical analysis been performed appropriately and rigorously? 

Reviewer #1: No

Reviewer #2: Yes

3. Have the authors made all data underlying the findings in their manuscript fully available?

Reviewer #1: No

Reviewer #2: Yes

4. Is the manuscript presented in an intelligible fashion and written in standard English?

Reviewer #1: Yes

Reviewer #2: Yes

5. Review Comments to the Author

Reviewer #1: The review of the paper entitled “Neural network-based method to stratify people at risk for developing diabetic foot: a support system for health professionals” Background and Objective:Diabetes Mellitus (DM) is a chronic disease with a high prevalence worldwide. Diabetic foot is one of the DM complications and compromises health and quality of life, due to the risk of lower limb amputation. This work is aimed to build a risk classification system for the development of the diabetic foot, using Artificial Neural Networks (ANN). Methods: It is a methodological study in which two databases were used, one for system design (training and validation) containing 250 people with DM and another for testing, containing 141 people. Each person answered a questionnaire with 54 questions about foot care and sociodemographic information. People from both databases were classified by specialists as high or low risk for diabetic foot. Supervised ANN (multi-layer Perceptron - MLP) models were exploited and a smartphone app was built. The app returns a personalized report indicating self-care for each user. System Usability Scale (SUS) was used for the usability evaluation.

I have found the following concerns which need to correct…

1. The introduction part is in multiple paragraphs, please rearrange the article text. Arrange multiple paragraphs in one if applicable.

2. I never found any novel work that is highlighted.

3. Its confusion for readers that what is the data in your article? Still authors talking about the patients, please make sure that which kind of data are you using?

4. ANN is a very much older method, if authors try to apply any deep learning method, it could be helpful.

5. Authors only talking about the ANN is different forms, and try to change the dataset but not the original method. But still readers are unaware of data, what is the input to the system?

6. Figure one and other figures are not according to the standard. It should be 300dpi.

7. Please add more recent papers in the references.

8. Try to compare your work with others in the same domain.

Reviewer #2: The manuscript focuses on detecting Diabetes Mellitus (DM) by using AI technique. Reducing the contributory reasons of DM from 54 to 10, appears to be a contribution of this paper. Although, with the available compute power today, processing 54 attributes for class identification is not a challenge, however, from an end-user perspective, data collection against 10 questions would be more efficient. As a suggestive added check, identification of most important parameters in low risk categorise may be re-validated. The quality of images used in the paper also needs improvement for enhanced visibility.

The manuscript is written nicely and presents an important issue related to healthcare. The paper is recommended for approval.

6. PLOS authors have the option to publish the peer review history of their article (what does this mean?). If published, this will include your full peer review and any attached files.

Reviewer #1: No

Reviewer #2: **Yes: **Raye Mahmood Ahmad

---

## [Author Response · Author response to Decision Letter 0]

3 Jan 2023

Reviewer #1: 

The review of the paper entitled “Neural network-based method to stratify people at risk for developing diabetic foot: a support system for health professionals” Background and Objective: Diabetes Mellitus (DM) is a chronic disease with a high prevalence worldwide. Diabetic foot is one of the DM complications and compromises health and quality of life, due to the risk of lower limb amputation. This work is aimed to build a risk classification system for the development of the diabetic foot, using Artificial Neural Networks (ANN). Methods: It is a methodological study in which two databases were used, one for system design (training and validation) containing 250 people with DM and another for testing, containing 141 people. Each person answered a questionnaire with 54 questions about foot care and sociodemographic information. People from both databases were classified by specialists as high or low risk for diabetic foot. Supervised ANN (multi-layer Perceptron - MLP) models were exploited and a smartphone app was built. The app returns a personalized report indicating self-care for each user. System Usability Scale (SUS) was used for the usability evaluation.

I have found the following concerns which need to correct…

1. The introduction part is in multiple paragraphs, please rearrange the article text. Arrange multiple paragraphs in one if applicable.

RESPONSE: Thank you for this suggestion. We have rearranged the introduction section in which some paragraphs were arranged in one. 

2. I never found any novel work that is highlighted.

RESPONSE: We have rewritten this sentence in the new manuscript version. 

3. Its confusion for readers that what is the data in your article? Still authors talking about the patients, please make sure that which kind of data are you using?

RESPONSE: Our database consisted of 54 pieces of information from 250 people with diabetes. These data were collected via questionnaire in a health institution in a city in the state of Minas Gerais, Brazil. The instrument (questionnaire) used for data collection covered the risk factors for the development of diabetic foot proposed by the International Consensus on Diabetic Foot, and by the Primary Care Booklet to Assist People with DM from the Ministry of Health of Brazil, which resulted in a total of 54 variables. Variables related to people's self-care habits concerning their health, foot care, perceived changes in the feet, as well as sociodemographic and socioeconomic variables were included.

This information is in the first two paragraphs of Section 2. To make this part clearer, we rewrote the first paragraph. 

4. ANN is a very much older method, if authors try to apply any deep learning method, it could be helpful.

RESPONSE: In this paper we used a multi-layer perceptron (MLP), which is the most popular type of artificial neural network (ANN). An MLP is an ANN capable of handling both linearly separable and non-linearly separable data. It belongs to a class of neural networks known as feed-forward neural networks, which connect the neurons in one layer to the next layer in a forward manner. It consists of interconnected neurons which process data through three or more layers. The basic structure of an MLP consists of an input layer, one or more hidden layers and an output layer. There is no restriction on the number of hidden layers, however, an MLP usually has a small number of hidden layers. In our work we used only one hidden layer. Due to their simplicity, MLPs usually require short training times to learn the representations in data and produce an output. 

A Deep Neural Network (DNN) is simply an artificial neural network with deep layers. Deep layers in this context mean that the network has several layers stacked together for processing and learning from data. It is important to note that an MLP is considered an example of DNNs. Due to their complex nature, DNNs usually require long periods of time to train the network on the input data. Additionally, they require powerful computers with specialized processing units such as Tensor Processing Units (TPU) and Neural Processing Units (NPU).

It is possible that the use of a DNN instead of the simple MLP employed may lead to a model with slightly higher accuracy. On the other hand, it will require more data to be trained, a specialized processing unit, and probably would result in a more complex model. As we discussed in Section 5.1, we opted by more parsimonious models, and, in this case, the MLP is more appropriated than a DNN. 

5. Authors only talking about the ANN is different forms, and try to change the dataset but not the original method. But still readers are unaware of data, what is the input to the system?

RESPONSE: To make the data description better, we have rewritten section 2 so that more details about the variables and their codification was added in the new manuscript version. 

In addition, we included paragraph 4 in Section 4 of the new manuscript version, where we described the MLP inputs for each approach developed. 

6. Figure one and other figures are not according to the standard. It should be 300dpi.

RESPONSE: The figures were edited accordingly.

7. Please add more recent papers in the references.

RESPONSE: He revised the paper and included seven new references of the last two years. 

8. Try to compare your work with others in the same domain.

RESPONSE: In Section 5.2, Table 3, of the submitted manuscript version, we have already presented a comparison between our best model and a Competitive Neural Layer model proposed in reference [11] for the same purpose. In this case, the model proposed in [11] was implemented and applied to the same testing data set. In addition, in the Discussion subsection we have related our achievements with other works in the same domain. 

Reviewer #2: 

The manuscript focuses on detecting Diabetes Mellitus (DM) by using AI technique. Reducing the contributory reasons of DM from 54 to 10, appears to be a contribution of this paper. Although, with the available compute power today, processing 54 attributes for class identification is not a challenge, however, from an end-user perspective, data collection against 10 questions would be more efficient. As a suggestive added check, identification of most important parameters in low risk categorise may be re-validated. The quality of images used in the paper also needs improvement for enhanced visibility.

RESPONSE: Thank you for your positive comments and suggestions. We have included a sentence in paragraph five of the new manuscript version to highlight the contribution of the paper in making the data collection easier for the end-user by reducing the contributory reasons of the diabetic foot risk. 

Regarding the identification of most important parameters in low risk group, we have rewritten the last paragraph of Section 5.2 to make this part clearer. 

The manuscript is written nicely and presents an important issue related to healthcare. The paper is recommended for approval. 

RESPONSE: Thank you for the positive comments.

---

## [Decision Letter · Decision Letter 1]

21 Feb 2023

PONE-D-22-26803R1Neural network-based method to stratify people at risk for developing diabetic foot: a support system for health professionalsPLOS ONE

Dear Dr. Ferreira,

Thank you for submitting your manuscript to PLOS ONE. After careful consideration, we feel that it has merit but does not fully meet PLOS ONE’s publication criteria as it currently stands. Therefore, we invite you to submit a revised version of the manuscript that addresses the points raised during the review process.

We look forward to receiving your revised manuscript.

Kind regards,

Imran Ashraf, Ph.D.

Academic Editor

PLOS ONE

Journal Requirements:

Reviewers' comments:

Reviewer's Responses to Questions

**Comments to the Author**

1. If the authors have adequately addressed your comments raised in a previous round of review and you feel that this manuscript is now acceptable for publication, you may indicate that here to bypass the “Comments to the Author” section, enter your conflict of interest statement in the “Confidential to Editor” section, and submit your "Accept" recommendation.

Reviewer #1: All comments have been addressed

Reviewer #3: All comments have been addressed

2. Is the manuscript technically sound, and do the data support the conclusions?

Reviewer #1: Yes

Reviewer #3: Yes

3. Has the statistical analysis been performed appropriately and rigorously? 

Reviewer #1: (No Response)

Reviewer #3: Yes

4. Have the authors made all data underlying the findings in their manuscript fully available?

Reviewer #1: Yes

Reviewer #3: Yes

5. Is the manuscript presented in an intelligible fashion and written in standard English?

Reviewer #1: Yes

Reviewer #3: (No Response)

6. Review Comments to the Author

Reviewer #1: Review of the article entitled “Neural network-based method to stratify people at risk for developing diabetic foot: a support system for health professionals”. Diabetes Mellitus (DM) is a chronic disease with a high 16 prevalence worldwide. Diabetic foot is one of the DM complications and compromises 17 health and quality of life, due to the risk of lower limb amputation. This work is aimed 18 to build a risk classification system for the development of the diabetic foot, using

19 Artificial Neural Networks (ANN).

The article has been prepared according to my suggestion. Now have the following minor concerns.

1. Thoroughly check the grammatical errors.

2. Add some references in the background study which improve your paper’s literature review, for example…

10.1109/ACCESS.2019.2896961

10.1109/ACCESS.2021.3056516

https://doi.org/10.3390/app9010069

https://doi.org/10.1155/2022/7954333

https://doi.org/10.3390/math10050796

https://doi.org/10.1155/2022/4942637

Reviewer #3: In this paper, the authors are proposed “Neural network-based method to stratify people at risk for developing diabetic foot: a support system for health professionals

”.

The strengths of the paper are that it is well structured, the description of the related work is well done and that results are extensively compared to results of the similar research.

The authors addressed all the comments. So, i recommend to accept this manuscript.

7. PLOS authors have the option to publish the peer review history of their article (what does this mean?). If published, this will include your full peer review and any attached files.

Reviewer #1: No

Reviewer #3: No

---

## [Author Response · Author response to Decision Letter 1]

18 Mar 2023

Response to reviewer #1

We carefully reviewed the manuscript.

The suggested references are related to image processing and segmentation methods based on Deep Learning approaches. Thus, we decided to reference these on the conclusion section where we described our intention for future works. 

Response to Reviewer #3

Thank you for the positive comments.

---

## [Decision Letter · Decision Letter 2]

20 Apr 2023

PONE-D-22-26803R2Neural network-based method to stratify people at risk for developing diabetic foot: a support system for health professionalsPLOS ONE

Dear Dr. Ferreira,

Thank you for submitting your manuscript to PLOS ONE. After careful consideration, we feel that it has merit but does not fully meet PLOS ONE’s publication criteria as it currently stands. Therefore, we invite you to submit a revised version of the manuscript that addresses the points raised during the review process.

We look forward to receiving your revised manuscript.

Kind regards,

Imran Ashraf, Ph.D.

Academic Editor

PLOS ONE

Journal Requirements:

Reviewers' comments:

Reviewer's Responses to Questions

**Comments to the Author**

1. If the authors have adequately addressed your comments raised in a previous round of review and you feel that this manuscript is now acceptable for publication, you may indicate that here to bypass the “Comments to the Author” section, enter your conflict of interest statement in the “Confidential to Editor” section, and submit your "Accept" recommendation.

Reviewer #1: (No Response)

2. Is the manuscript technically sound, and do the data support the conclusions?

Reviewer #1: Partly

3. Has the statistical analysis been performed appropriately and rigorously? 

Reviewer #1: Yes

4. Have the authors made all data underlying the findings in their manuscript fully available?

Reviewer #1: Yes

5. Is the manuscript presented in an intelligible fashion and written in standard English?

Reviewer #1: Yes

6. Review Comments to the Author

Reviewer #1: Review of the manuscript entitled “Neural network-based method to stratify people at risk for developing diabetic foot: a support system for health professionals”. Diabetes Mellitus (DM) is a chronic disease with a high prevalence worldwide. Diabetic foot is one of the DM complications and compromises health and quality of life, due to the risk of lower limb amputation. This work is aimed to build a risk classification system for the development of diabetic foot, using Artificial Neural Networks (ANN). Methods: It is a methodological study in which two databases were used, one for system design (training and validation) containing 250 people with DM and another for testing, containing 141 people. Each person answered a questionnaire with 54 questions about foot care and socio-demographic information. People from both databases were classified by specialists as high or low risk for the diabetic foot. Supervised ANN (multi-layer Perceptron - MLP) models were exploited and a smartphone app was built. The app returns a personalized report indicating self-care for each user. The system Usability Scale (SUS) was used for the usability evaluation. Results:

I have read the updated paper and still I found some issues which need to be corrected.

1. Discussion is poor, please add a table in the discussion section that should compare your results with other authors in the same domain.

2. Please add more recent medical and deep learning research papers in the background section.

https://doi.org/10.1109/ACCESS.2019.2896961

https://doi.org/10.1109/ACCESS.2021.3056516

https://doi.org/10.3390/app9010069

https://doi.org/10.1155/2022/7954333

https://doi.org/10.3390/math10050796

https://doi.org/10.1155/2022/4942637

3. Still have a lot of grammatical errors.

7. PLOS authors have the option to publish the peer review history of their article (what does this mean?). If published, this will include your full peer review and any attached files.

Reviewer #1: No

---

## [Author Response · Author response to Decision Letter 2]

4 Jun 2023

Answers to the Reviewers 

Reviewer #1: Review of the manuscript entitled “Neural network-based method to stratify people at risk for developing diabetic foot: a support system for health professionals”. Diabetes Mellitus (DM) is a chronic disease with a high prevalence worldwide. Diabetic foot is one of the DM complications and compromises health and quality of life, due to the risk of lower limb amputation. This work is aimed to build a risk classification system for the development of diabetic foot, using Artificial Neural Networks (ANN). Methods: It is a methodological study in which two databases were used, one for system design (training and validation) containing 250 people with DM and another for testing, containing 141 people. Each person answered a questionnaire with 54 questions about foot care and socio-demographic information. People from both databases were classified by specialists as high or low risk for the diabetic foot. Supervised ANN (multi-layer Perceptron - MLP) models were exploited and a smartphone app was built. The app returns a personalized report indicating self-care for each user. The system Usability Scale (SUS) was used for the usability evaluation. Results:

I have read the updated paper and still I found some issues which need to be corrected.

1. Discussion is poor, please add a table in the discussion section that should compare your results with other authors in the same domain.

Authors: Thank you for your comment. We have included a further discussion at the end of Section 5.1, in which we considered recent works using machine learning approaches to access diabetic foot development. In addition, we have included Table 4 in the revised manuscript, which presents a comparison of methods in terms of feature selection and classification techniques, sample size, clinical exams, number of available variables, number of selected variables, and performance. 

2. Please add more recent medical and deep learning research papers in the background section.

https://doi.org/10.1109/ACCESS.2019.2896961

https://doi.org/10.1109/ACCESS.2021.3056516

https://doi.org/10.3390/app9010069

https://doi.org/10.1155/2022/7954333

https://doi.org/10.3390/math10050796

https://doi.org/10.1155/2022/4942637

Authors: Thank you for your suggestion, these are interesting papers. We have included more recent medical and deep learning research papers in the Introduction section. 

3. Still have a lot of grammatical errors

Authors: We are very sorry for missing these grammatical errors. We sent the revised manuscript for a professional grammar review and hope that there is no further error in this regard.

---

## [Decision Letter · Decision Letter 3]

28 Jun 2023

Neural network-based method to stratify people at risk for developing diabetic foot: a support system for health professionals

PONE-D-22-26803R3

Dear Dr. Danton Ferreira, PhD,

We’re pleased to inform you that your manuscript has been judged scientifically suitable for publication and will be formally accepted for publication once it meets all outstanding technical requirements.

Kind regards,

Melissa Orlandin Premaor, M.D., Ph.D

Academic Editor

PLOS ONE

Reviewers' comments:

Reviewer's Responses to Questions

**Comments to the Author**

1. If the authors have adequately addressed your comments raised in a previous round of review and you feel that this manuscript is now acceptable for publication, you may indicate that here to bypass the “Comments to the Author” section, enter your conflict of interest statement in the “Confidential to Editor” section, and submit your "Accept" recommendation.

Reviewer #1: All comments have been addressed

Reviewer #3: All comments have been addressed

2. Is the manuscript technically sound, and do the data support the conclusions?

Reviewer #1: Yes

Reviewer #3: Yes

3. Has the statistical analysis been performed appropriately and rigorously? 

Reviewer #1: Yes

Reviewer #3: Yes

4. Have the authors made all data underlying the findings in their manuscript fully available?

Reviewer #1: Yes

Reviewer #3: Yes

5. Is the manuscript presented in an intelligible fashion and written in standard English?

Reviewer #1: Yes

Reviewer #3: Yes

6. Review Comments to the Author

Reviewer #1: The review of the article entitled “Diabetes Mellitus (DM) is a chronic disease with a high

worldwide prevalence. Diabetic foot is one of the DM complications and compromises

health and quality of life, due to the risk of lower limb amputation. This work aimed to

build a risk classification system for the evolution of diabetic foot, using Artificial

Neural Networks (ANN). Methods: This methodological study used two databases, one

for system design (training and validation) containing 250 participants with DM and

another for testing, containing 141 participants. Each subject answered a questionnaire

with 54 questions about foot care and sociodemographic information. Participants from

both databases were classified by specialists as high or low risk for diabetic foot.

Supervised ANN (multi-layer Perceptron - MLP) models were exploited and a

smartphone app was built.

Now the paper is written and organized well. I have no new questions to ask, all of my suggestions are fulfilled by the authors.

Reviewer #3: The research meets all applicable standards for the ethics of experimentation and research integrity. So i recommend to accept the research

7. PLOS authors have the option to publish the peer review history of their article (what does this mean?). If published, this will include your full peer review and any attached files.

Reviewer #1: No

Reviewer #3: No

---

## [Editor Report · Acceptance letter]

3 Jul 2023

PONE-D-22-26803R3 

Neural network-based method to stratify people at risk for developing diabetic foot: a support system for health professionals 

Dear Dr. Ferreira:

I'm pleased to inform you that your manuscript has been deemed suitable for publication in PLOS ONE. Congratulations! Your manuscript is now with our production department. 

Kind regards, 

on behalf of

Dr. Melissa Orlandin Premaor 

Academic Editor

PLOS ONE